# Within-subjects ultra-short sleep-wake protocol for characterising circadian variations in retinal function

Hannah Sophie Heinrichs[1,2], Manuel Spitschan[1,2,3]*

**1** Max Planck Institute for Biological Cybernetics, Max Planck Research Group Translational Sensory and Circadian Neuroscience, Tübingen, Germany, **2** Technical University Munich, TUM School of Medicine and Health, Munich, Germany, **3** Technical University Munich, TUM Institute for Advanced Study (TUM-IAS), Garching, Germany

* manuel.spitschan@tum.de

**Data availability statement:** Upon conclusion of the primary analyses, the data will be made available under the Creative Commons licence (CC-BY) with no reservations in the supplementary material of the research publication and/or on a public repository (e.g., FigShare). Pseudonymised research data will be transferred once data collection is completed and conclusions of the main hypotheses have been submitted.

## Abstract

Prior studies suggest that visual functions undergo time-of-day variations. Under naturalistic entrainment, diurnal changes in physiology may be driven by circadian and/or homeostatic processes, and repeated measurements at different times of day are thus not suitable to draw unambiguous conclusions about circadian effects on visual function. In this study, we disentangle circadian and homeostatic effects on variations of retinal function. We examine the earliest stages of image-forming (temporal contrast sensitivity of the post-receptoral channels) and non-image forming visual functions (pupillary light response) by employing a short forced-desynchrony multiple-naps protocol lasting 40 hours. Participants (n = 12, 50% female) will stay in a controlled time-isolating environment under dim-light conditions and adhere to an ultra-short sleep-wake cycle, alternating between 2h30m of wake time in dim light and hour of sleep in no light. During eleven intervals of wakefulness, participants will undergo psychophysical and pupillometric assessments with silent-substitution stimuli. We hypothesize that the sensitivity of retinal mechanisms undergoes circadian variations. This hypothesis will be investigated by separately determining psychophysical contrast thresholds to silent-substitution stimuli targeting the post-receptoral (consistency) pathways (isoluminant red-green, L–M; isoluminant blue-yellow, S; luminance, L+M+S). We will furthermore measure the pupillary light response to peripheral stimuli (annulus 10°–30°) in comparison to the response to stimuli isolating or including melanopsin stimulation. All stimuli will be delivered at constant retinal irradiance using a Maxwellian view system or artificially restricting pupil size. Additionally, we will quantify and report effects of our test stimuli on the circadian system by comparing the dim light melatonin onset (DLMO) timing during two supplementary evening sessions, comparing dim-light conditions to such with experimental light exposure. Our work informs the fundamental biological mechanisms underlying the influence of light on the human circadian system. Based on our findings, current models about the sensitivity of the circadian system may need to be modified in order to account for the bidirectional influence of circadian function and photoreception.

**Funding:** This research was supported by the Max Planck Society. The funder had no role in study design, data collection and analysis, decision to publish, or preparation of the manuscript.

**Competing interests:** The authors have declared that no competing interests exist.

# 1 Introduction

## 1.1 Processing of light in the retina

Exposure to light enables vision and visual perception, allowing us to see the world in its colourful and spatial detail [1]. Additionally, light exposure also influences our physiology and behaviour through circadian synchronization [2], melatonin suppression [3–6] and modulation of alertness [6].

The pathways underlying these distinct functions are also anatomically distinct and separated into image-forming and non-image-forming pathways. Signals in both pathways arise from the three classes of photoreceptors in the retina: rods, cones and the melanopsin-containing intrinsically photosensitive retinal ganglion cells (ipRGCs). Image-forming signals primarily stem from rods and three types of cones with different spectral sensitivity. Phototransduction takes place at the photoreceptor level, but before the resulting signal is passed on to the retinal ganglion cells and eventually subcortical (lateral geniculate nucleus; LGN) and cortical targets (primary visual cortex; V1), the signal is recombined through retinal wiring. The coding of luminance is based on the summation of L- and M-cone signals, and two opponent channels encode colour: the red-green channel that is driven by the antagonism of L-cone versus M-cone signals (L-M), and the yellow-blue channel driven by the S-cone versus the sum of L- and M-cone signals (S-[L+M]) [7]. The ipRGCs represent a subset of retinal ganglion cells which express their own photopigment, melanopsin. In addition to this intrinsic photosensitivity, they also receive synaptic input from the rods and cones.

The effects of light on human circadian and neuroendocrine physiology are orchestrated by the suprachiasmatic nucleus (SCN) in the hypothalamus. Through the retinohypothalamic pathway, which relays information from the retina to the hypothalamus, the SCN encodes light to synchronize the internal circadian rhythm to the external environment, rendering light input an important circadian signal [8]. Given the widespread influence of the central pacemaker in many physiological systems and tissues, and peripheral clocks in other tissue types, it is highly likely that one or all sites of the input pathway is also affected by a circadian modulation.

Across the day, light exposure has different and opposing circadian effects depending on the timing of exposure. Given by the phase response curve (PRC) to light, light exposure in the morning advances the circadian phase while evening light exposure will cause a delay [9–11]. To what extent the PRC is driven by a modulation of sensitivity in the retina or mechanisms along the retinohypothalamic pathway is unknown. Some studies have investigated how the pupillary light reflex (PLR) is modulated, as it reflects the response of all photoreceptors to light [12,13]. They revealed that there are time-of-day differences in melanopsin-mediated pupil responses [14,15], indicating that at least some variability arises and is measurable as early as in the retina. Indeed, if this were the case, i.e., that there is a circadian "gating" of the retinal signal, it could explain the effects seen downstream. Only a few studies have documented the diurnal variation of basic visual functions, such as luminance perception, including temporal contrast [16] and spatial contrast sensitivity [17]. Some studies have also investigated diurnal variations in colour vision [17,18]. Additionally, there is strong evidence for diurnal variations of interocular pressure and corneal thickness, which could lead to differences in retinal function [19,20].

However, the emerging pattern of results is somewhat heterogeneous, and does not factor in that time-of-day variations in retinal and physiology cannot be uniquely attributed to the circadian clock. The goal of this study is to characterise the variability of retinal functions as a function of circadian phase.

## 1.2 Disentangling circadian and homeostatic process

The intercorrelated nature of processes controlled by the circadian clock and behavioural rhythms poses considerable challenges to the investigation of uniquely circadian effects, requiring well-controlled studies spanning many hours to multiple weeks. Briefly, daily variation in human physiology, behaviour, and experience at the 24-hour scale are governed by two independent processes, an endogenous circadian rhythm and a sleep homeostat [21,22]. Under normal, entrained conditions, these processes are working in tandem to regulate the propensity for sleep, and to create regular cycles of sleep and wakefulness.

The two-process model postulates that the homeostatic "process S" gradually increases during periods of wakefulness, giving rise to an increasingly stronger drive for sleep the longer one stays awake, and affecting a variety of physiological and cognitive functions. During sleep, sleep pressure decreases exponentially, allowing for recovery and restoring balance before the next sleep-wake cycle. At the same time, the circadian "process C" regulates various physiological functions and signals, which in interaction with the sleep homeostat influence sleep timing and propensity. Under entrained conditions, the sleep-wake cycle is consequently aligned with the internal body clock and external environmental cues, making both processes correlated with one another [23].

In the laboratory, it is possible to disentangle their effects on neurobehavioural functions by systematically manipulating sleep timing, sleep duration and contextual factors such as ambient light, temperature and food intake. There are various within-person study designs that requiring repeated measurements of outcome parameters and careful monitoring and controlling of circadian signals and homeostatic influences [23–25].

## 1.3 Ultra-short sleep-wake forced-desynchrony protocol

Here, we will employ a 40-hour forced-desynchrony (FD), multiple-naps protocol with a night-day/light-dark (LD) cycle totalling 3h45m. In this protocol, sleep pressure is kept at a constant low level by rapidly alternating between sleep and wake periods (i.e. regular naps). Other homeostatic influences like food intake and activity will be adapted to the ultra-short rhythm. External influences including ambient temperature and dim light during wakefulness are kept constant to minimize the influence on the circadian rhythm.

In the literature, a similar protocol has been successfully implemented before, scheduling sleep and wake time in a 2:1 ratio of imposed day length of 3h45m [26,27]. 150 min (2h30m) wake intervals in dim light (<10 lux) provide opportunity for measurements and are followed by 75 min (1h15m) sleep intervals in darkness. The protocol lasts 40 hours, yielding 11 repeating light-dark (LD) cycles or measurement blocks.

This study will isolate circadian effects on early stages of visual processing from homeostatic effects with a short forced-desynchrony protocol. Participants will stay in the laboratory for 40 hours, which will cover, on average, 1.5 circadian periods.

They will adhere to an ultra-short sleep-wake rhythm schedule, comprised of 2h30m (150 minutes) of wake time and 1h15m (75 minutes) of sleep opportunity. During each scheduled instance of wakefulness, a series of (psycho-)physical parameters will be measured in order to determine sensitivity of photoreceptor mechanisms, circadian phase, sleep pressure, and other possible physiological outcomes assumed to undergo circadian change. Specifically, we will repeatedly collect data on: (1) psychophysical performance, and more specifically temporal contrast sensitivity (tCS) (2) the immediate maximum pupil constriction amplitude (pupillary light response, PLR) to carefully crafted light stimuli, (3) intra-ocular pressure, (4) structural properties of the cornea and retina (5) salivary melatonin concentration,

(6) mood, and (7) sleepiness; additionally, there will be continuous measurements of physiological functions (8) temperature in the core body and the proximal-distal gradient of skin temperature, (9) heart rate variability via electrocardiography (ECG), and (10) interstitial glucose concentration.

## 1.4 Hypotheses

Building upon prior evidence for a time-of day dependency in image-forming (i.e., luminance and colour perception) and non image-forming functions (i.e., pupillary light response), we hypothesize that photoreceptor sensitivity and retinal mechanisms are modulated by the circadian rhythm.

Our hypotheses regarding variations in non-image forming function investigate circadian effects in the pupillary light reflex to different conditions.

The detailed confirmatory hypotheses state that the PLR shows circadian effects in the response to

1. modulation of the colour opponent process L-M with constant activation of S and melanopsin (L–M) [H1a];
2. modulation of the S-cones with constant activation of L-cone, M-cone, and melanopsin activation (S) [H1b];
3. joint modulation of L-, M-, and S-cones (L+M+S) [H1c];
4. selective modulation of melanopsin activation (Mel, constancy of cone activation) [H1d].

To investigate circadian effects isolated from homeostatic effects in the image-forming pathway by evaluating temporal contrast threshold to two different frequencies, focusing on the following photoreceptor mechanisms using silent-substitution stimuli: activation of the luminance channel (L+M+S), post-receptoral red-green channel (L–M), activation of the S-cones (S-(L+M)), activation of melanopsin-driven signaling in ipRGCs. We will evaluate the contrast sensitivity to:

1. modulation of the colour opponent process L-M with constant activation of S (L–M⋆) [H2a];
2. modulation of the S-cones with constant activation of L-cones and M-cones (S⋆) [H2b];
3. joint modulation of L-, M- and S-cones (L+M+S⋆) [H2c],

with none of the conditions controlling for melanopsin modulation, as indicated by the asterisk. The hypotheses regarding psychophysical performance in image-forming vision, relates to two different frequencies, yielding six different hypothesis: each hypothesis will be investigated for a low frequency (2 Hz) and a high frequency (8 Hz) to identify and possible explore differences based on temporal sensitivity.

## 1.5 Prospective impact of the study

We will characterise circadian influences on photoreceptor sensitivity and sensitivity of post-receptoral mechanisms, thereby gaining further insights into mechanisms and extent of variations of visual performance. Studying circadian dependency in human visual function will deepen our mechanistic understanding of the link between image-forming and

non-image-forming functions. Furthermore, our results have practical implications for professional contexts that involve driving or rely on colour discrimination and thus require high visual performance, and is of particular importance when circumstances impose circadian disruption and sleep disruption, such as in the medicine, aviation or military service. Finally, from a clinical perspective, the localisation of circadian variation in eye physiology may also have implications for the treatment and management of eye conditions such as glaucoma.

## 2 Methods

### 2.1 Study sample

In this study, we will recruit and enrol twelve healthy participants aged 18–35 years (target 50% female) with no ocular and retinal diseases, no psychiatric or neurological diseases, with normal sleep-wake behaviour, normal colour vision, and matching all criteria listed in Table 1 and not matching any criteria listed in Table 2.

**2.1.1 Recruitment.** We will recruit participants using a multi-modal strategy using flyers, posters, mailing lists and other outlets, word-of-mouth and advertisements placed on the internet. The first point of contact will be an online screening. Upon completion of the online screening, their suitability for the study will be determined automatically, followed by an invitation or removal from the participant pool. Participants will undergo a thorough in-laboratory screening described in Sect 2.4.1. During screening, formal criteria for participation will be checked again with regard to their physical health and psychological stability. Participants will be informed about the study protocol and experimental procedures, and given opportunity to ask questions before giving written consent.

**2.1.2 Remuneration.** Participants will be remunerated for their time. There will be two in-laboratory evening sessions followed by an adaptation night in the lab (more details in Sect 2.4.2). For each of the evenings, participants will be compensated with €30 plus €20 for the adaptation nights. The second evening session and adaptation night will be immediately followed by the forced-desynchrony protocol starting in the morning. For the forced-desynchrony session of 40 hours, they will receive a compensation of €400. Participants may leave at any point upon request, or will be excluded in case of reduced compliance to an extent

**Table 1. Inclusion criteria.**

| Domain | Assessment method | Criterion | Time of Screening |
|---|---|---|---|
| Age | Self-report | ≥18 years <br> ≤35 years | Online screening |
| Physical health | Self-report | Good physical health | Online screening |
| Mental health | Self-report | Good mental health | Online screening |
| Ocular health | Ophthalmological examination | Good ocular health | In-person screening |
| Visual acuity | Landolt C | Normal or corrected-to-normal vision | In-person screening |
| Colour vision | HRR Pseudoisochromatic Plates (4th edition) | Normal colour vision | In-person screening |
| Sleep and circadian regularity | Self-report | Preference of high ranking participants | Online screening |
| Hormone status | Self-report | Women and men without any hormonal disorders; women with or without hormonal contraception | Online screening |

**Table 2. Exclusion criteria.**

| Domain | Assessment method | Criterion | Time of screening |
|---|---|---|---|
| Body mass index (BMI) | Calculated based on body weight in everyday clothes without shoes and height | <18.5 or >25 | In-lab screening |
| Substance abuse | Self-report, AUDIT | AUDIT >7 | Online screening |
| Depressive symptoms | Self-report | | In-lab screening |
| History of anxiety disorder | Self-report | | In-lab screening |
| Extreme chronotype | Self-report, MCTQ (MSF-SC) | <2:00 or >5:30 | Online screening |
| Medication use | Self-report | Any use of medication (except for hormonal contraception) | Online & in-lab screening |
| Smoking | Self-report | Habitual smoking | Online screening |
| Photosensitive epilepsy | Self-report | Diagnosis of epilepsy | Online & in-lab screening |
| Shift work | Self-report | No shift work in the past 3 months | Online screening |
| Transmeridian travel | Self-report | No inter-time zone travel >2 time zones in the past 3 months | Online screening |
| Pregnancy | Self-report, HCG urine test | Positive result | Online screening |
| Endocrine alterations | Self-report | | Online screening |
| Drug use | AMP, BZD, COC, MOR/OPI, THC pane | Positive result | Light exposure session & forced desynchrony session |
| Alcohol use | Breathalyser | >0.0% | Light exposure session & forced desynchrony session |

that renders the data deficient. When returning the actigraph, participants will be rewarded €20. As participants are incentivised to provide complete data for actimetry and the sleep diary, they will be rewarded an additional €80 if less than 20% of data is missing. The maximum remuneration is therefore €600.

**2.1.3 Sample size.** This study will follow a within-person design. Each participant provides eleven repeated measurements. We will determine evidence for our hypotheses using a sequential Bayesian sampling strategy, such that data collection will continue until evidence strength suffices, or until a maximum of twelve participants. This maximum sample size is set by resource limitations. For the sequential sampling strategy, we will use Sequential Bayes Factor (BF) of the statistical model testing circadian modulation of melanopsin-dependent pupil constriction. The stopping criterion is BF >10 or BF <1/10 to quantify evidence strength for or against a circadian effect.

## 2.2 Study design

The study is set up to be a within-participant design. We will invite one participant at a time. Each participant will take part in one evening session a week after starting a circadian stabilisation phase, then come in for another evening session that is followed by an adaptation night and the subsequent completions of the forced-desynchrony protocol, which encompasses 11 measurement blocks and spans 40 hours. Each measurement block encompasses various measurements. The order of measurements is constant between participants. In the pupillometric measurements, experimental conditions, i.e. direction of post-receptoral stimulation, will be counterbalanced between participants, while for the psychometric measurements, the order of experimental conditions will be fixed, while frequency conditions are randomised.

Repeating measurement blocks are interspersed with sleep opportunities to prevent sleep deprivation, thus systematically shortening a typical sleep-wake cycle from approximately 24 hours to 3h45m. During one period of 24 hours, participants will thus complete 6.4 sleep-wake cycles. Additionally, throughout the entire study period, participants will provide a combination of regular ambulatory measurements and experience sampling starting up to two weeks before the measurements. Our study design as shown in Fig 1 will yield longitudinal data spanning up to three weeks.

## 2.3 Study duration & timeline

Each person participating in the study will be enrolling for a total of three weeks, consisting of continuous field measurements and one in-laboratory screening and two separate visits for control and experimental sessions.

After getting in touch with the study team, participants will first complete an online screening via the online platform REDCap [28,29]. The data from the online questionnaires or experience sampling is submitted to a REDCap server that is set up and maintained by the Chronobiology & Health team at TUM. It is set up as a virtual machine hosted by the Leibniz-Rechenzentrum der Bayerischen Akademie der Wissenschaften. If their answers comply with the inclusion and exclusion criteria, they will be invited for an in-person screening and, upon inclusion in the study, for two in-laboratory visits to take part in experiments. For an overview of the study participation, see Fig 1.

The online screening will be complemented by an in-person screening by a psychologist to confirm eligibility for the study. Since the forced-desynchrony protocol places participants in

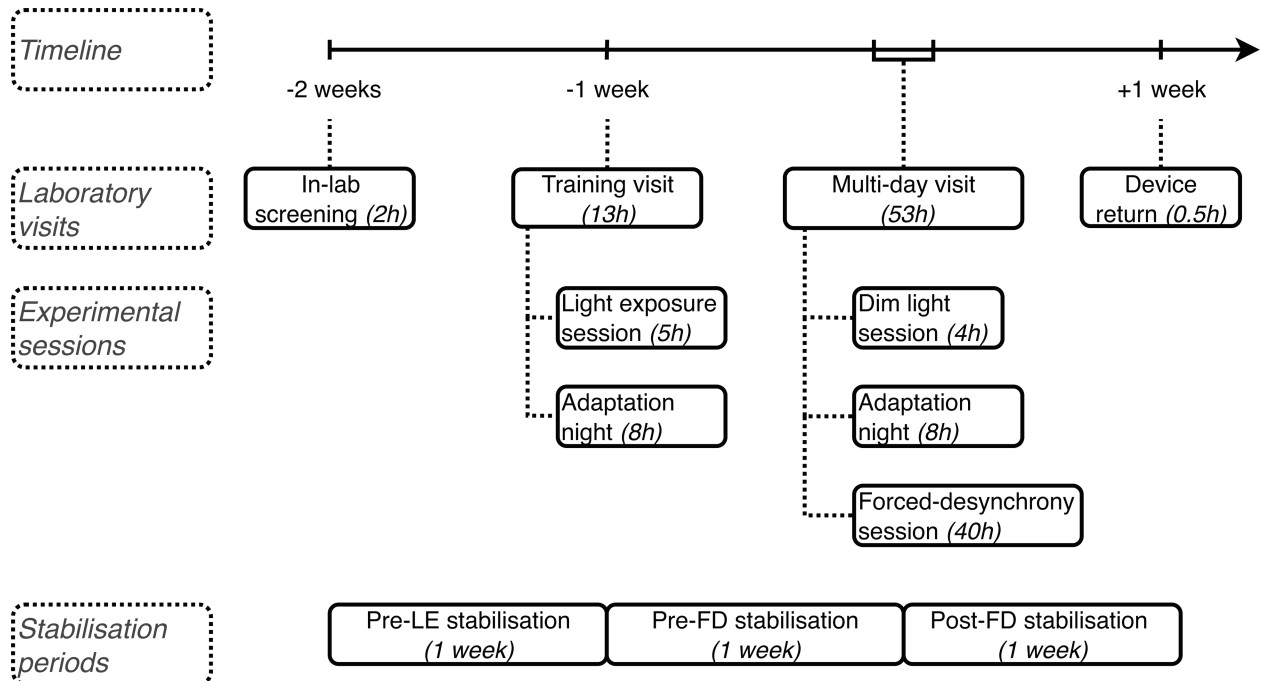

**Fig 1. Timeline of the study protocol.** The figure illustrates the sequence of circadian stabilisation periods, in-laboratory visits, and experimental sessions over a three-week period. Key events are shown relative to the onset of the dim light session. In-laboratory visits can include nights of sleep for adaptation and multiple sessions; stabilisation happens continuously throughout the study, but are labelled in relation to in-laboratory visits.

an exceptional environment, our inclusion criteria, the screening process, and procedures to ensure continued welfare are guided by suggestions from literature [30,31].

Upon inclusion in the study, participants will start a two-week long circadian stabilisation period. They will be invited for a first evening experimental session, the light exposure session, one week into the circadian stabilisation and one week before the forced-desynchrony session. Visual tests will be administered throughout the evening that achieve an experimental light exposure similar to the tasks administered during a wake period of within the forced-desynchrony session. The light exposure session also provides opportunity for the participant to get accustomed to the measurement instruments and prevent novelty or perceptual learning during the main study. After the evening session, participants will stay in the lab for a full night of sleep as their first adaptation night. Circadian outcomes from this session, including dim-light melatonin onset (DLMO) and core body temperature (CBT), as well as vigilance tests will be compared to those of an evening session under dim-light conditions. The purpose of this comparison is to quantify the potency of the light exposure during the protocol to elicit a circadian phase shift (for more information, see Sect 2.4.2).

The second laboratory visit starts with one evening session under dim-light conditions and another adaptation night when participants will have a full night of sleep (8 hours), followed by the forced-desynchrony session that involves 40 hours on a ultra-short sleep-wake cycles. Participants will enter the lab five hours before their habitual bedtime and complete a four-hour dim light session. The forced-desynchrony session starts 60 minutes after the end of the adaptation night. During the first 60 minutes, people have opportunity for personal hygiene and will be equipped with the sensors. 40 hours will be spent in constant environmental conditions, with a temperature of 20°C, and no time cues are provided. The main room is lit with dim light at <10 lux, so that light exposure is constant except during light exposure due to visual tests and during sleep. For an overview of the study period, see Fig 1.

The first ultra-short sleep-wake will start with a wake period. During wake time, all other tests will be carried out while sitting. To reach the devices, participants have to walk up within the laboratory for up to 30 meters. During wake time, light isocaloric snacks and water that are heated up to body temperature will be provided at three opportunities, mirroring breakfast, lunch, and dinner. After experiments, a remaining wake time of 30 minutes can be spent freely. The 2h30m wake period is followed by a 1h15m sleep period, that will be spent in a recumbent position. Participants will put on a light-tight sleep mask.

An overview timeline during the forced-desynchrony session is shown in Fig 2, and a more detailed timeline of single measurement blocks in Fig 3.

## 2.4 Experimental procedure

All devices were purchased from internal funds of the investigator and were not sponsored by the manufacturer(s).

**2.4.1 Screening.** An online and an in-laboratory screening will proceed the forced-desynchrony session. Participants will be included or excluded based on Table 1 and Table 2, respectively.

*Online screening.* In the online screening, participants will be asked for their age, biological sex and gender identity, and employment status. They will fill out questions on whether they are taking any medication, are smoking, have a diagnosis of epilepsy, and female-born participants will be asked about their reproductive status and menstrual cycle based on the adapted Reproductive Status Questionnaire [32]. Additionally, participants will complete a series of psychological questionnaires, delivered via the online platform REDCap. Participants

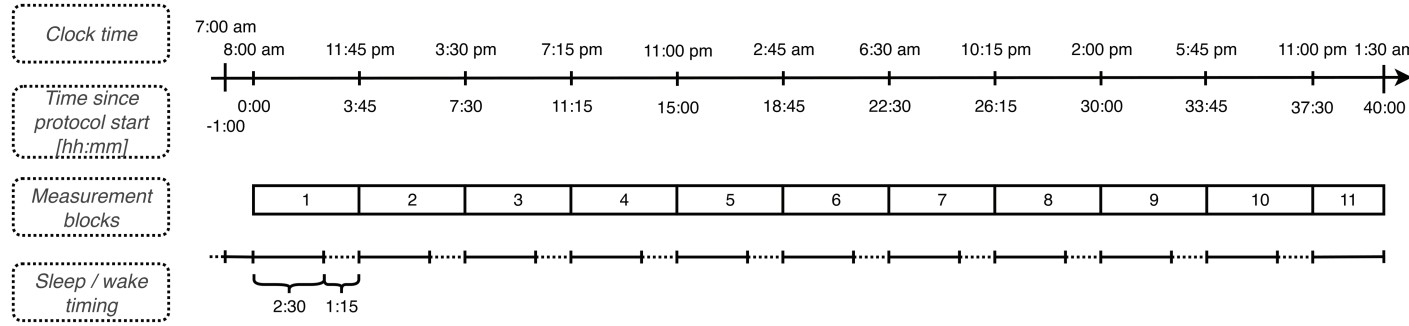

**Fig 2. Schematic of the 40-hour forced-desynchrony protocol.** This figure illustrates the ultra-short sleep-wake rhythm, displaying the relationship between clock time, time spent in the lab since protocol start, measurement blocks, and sleep-wake cycles. The timeline begins after an adaptation night and includes a 1-hour transition period for sensor placement. Each wake period (solid lines) lasts 2h30m, followed by a 1h15m sleep period (dashed lines).

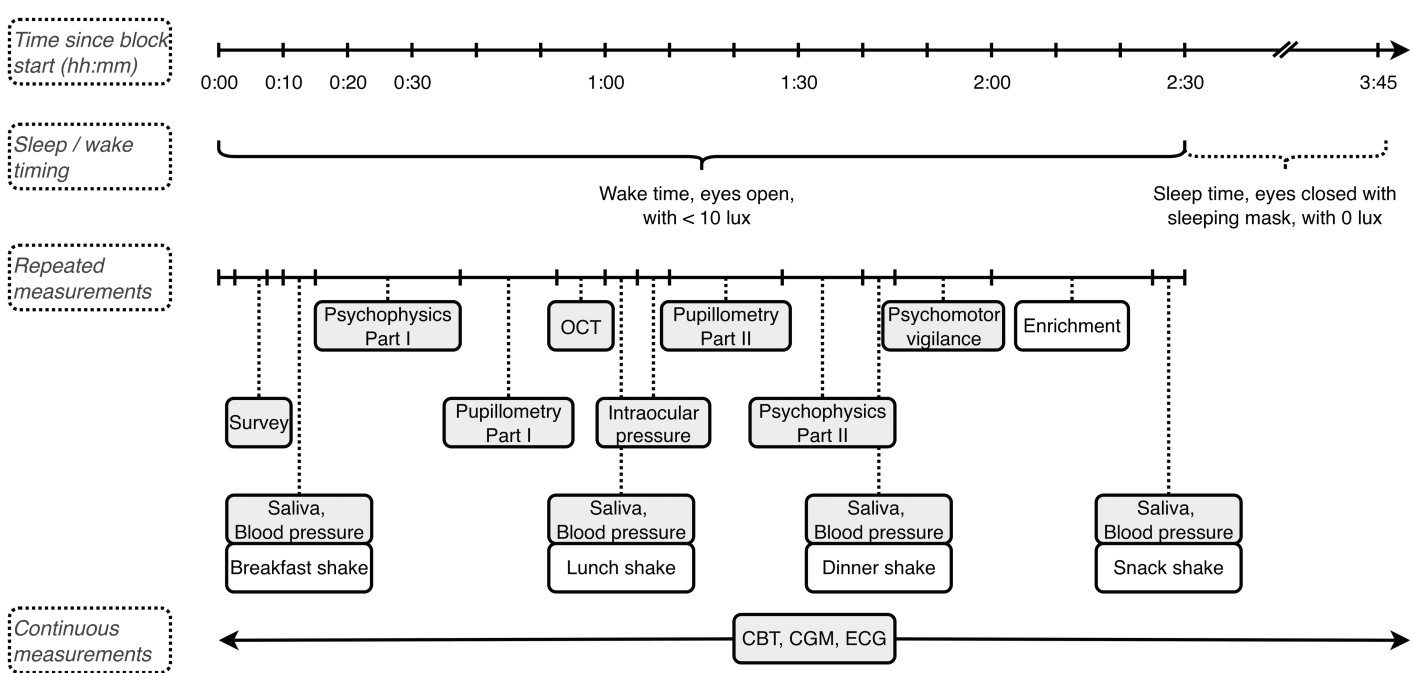

**Fig 3. Detailed timeline of a single measurement block during the 40-hour forced-desynchrony protocol.** This figure outlines the sequence and duration of measurements (grey background) and activities (white background) within one 2h30m wake period, including psychophysical tests, pupillometry, physiological measurements, and scheduled meal times. The continuous measurements, core body temperature (CBT), continuous glucose monitoring (CGM), and electrocardiography (ECG), are indicated at the bottom.

will complete the Alcohol Use Disorders Identification Test (AUDIT), an instrument to examine substance and alcohol abuse, in the self-report version [33]. As an assessment of sleep habits, we will use the Pittsburgh Sleep Quality Index (PSQI), an instrument to assess quality of sleep based on usual sleep habits and sleep disturbances [34], and the Epworth Sleepiness Scale (ESS) is used to assess habitual sleepiness [35]. The Munich Chronotype Questionnaire (MCTQ) will be used to determine chronotype [36]. It determines habitual midpoint of sleep, corrected for sleep duration on free days ($MSF_{SC}$). Only slightly early to slightly late chronotypes are admissible for reasons of scheduling group-wise participation [37]. Furthermore,

**Table 3. Measurement devices.**

| Device | Manufacturer | Purpose | Certification |
|---|---|---|---|
| AMPBpro | SOMNOmedics, Randersacker, DE | Blood pressure measurements | |
| Auto Non-Contact Tonometer NT-2000 | NIDEK CO., LTD., JP | Measurement of intra-ocular pressure | |
| e-Celsius | BodyCap, Hérouville-Saint-Clair, FR | Measurement of core body temperature using an ingestible pill | CE |
| FreeStyle Libre 3 | Abbott Diabetes Care Ltd., UK | Continuous glucose monitoring | |
| iButton | Maxim/Dallas Semiconductor Corp., USA | Peripheral body temperature measurements | |
| Metropsis Visual Function Assessment | Cambridge Research Systems, Rochester, UK | Psychophysical measurements | |
| NVBL | Martial Geiser, CH | Visual stimulation and pupillometry | |
| OCT2000 | TOPCON Corporation, JP | Ocular physiology measurements, incl. corneal thickness measurements and fundus photographs | CE |
| PVT-192 | Ambulatory Monitoring Inc., USA | Measurement of alertness | |

we will administer the Sleep Regularity Questionnaire (SRQ) and will preferably select people with regular sleep patterns over those with more irregular patterns [38].

*In-person screening.* During the in-person screenings, participants' physical health will be assessed using self-report questions, along with an examination by study personnel. The specific procedures encompass screening of their visual acuity using a Landolt C acuity test delivered on the Cambridge Research Systems Metropsis system, and their colour vision using HRR Pseudoisochromatic Plates. An ophthalmological exam including fundus photography and Optical Tomography images (macular image and optic disc image) will take place to confirm ocular and retinal health. The primary concern is to exclude individuals with any ocular disease or compromised visual information transmission, such as in glaucoma patients. A cup-to-disc ratio of >0.3 will lead to exclusion of the participant, as well as an intra-ocular pressure (IOP) >20 mmHg. Criteria are based on the European Guidelines for glaucoma diagnostics by the European Glaucoma Society to exclude participants with potentially diseased eyes [39]. These criteria may be conservative as higher cup-to-disc ratio does not necessarily indicate disease, especially for individuals with larger papilla size as then, automatically, the physiological papilla excavation increases. However, these criteria can be applied algorithmically, facilitating a faster decision without additional consultation. Eye dominance is determined using the Miles test [40]. In this test, a distant target is aligned binocularly through a small opening formed by both hands stretched away from the body and held steady. Then, each eye is alternatingly closed and opened to determine which monocular view of the target, seen through the aperture, matches the binocular view.

*Alcohol and drug compliance.* At the beginning of the first experimental session, the light exposure session, abstinence from alcohol and drugs will be verified using a breathalyser (Alkoholtester, ACE GmbH, Freilassing, Germany) to determine blood alcohol content (BAC), and a urine sample to determine AMP/BZD/COC/MOR/OPI/THC content (nal von minden GmbH, Moers, Germany). Participants will be asked to produce a urine sample in a collection device in a bathroom on-site. Experimenters will carry out the drug test

by immersing a test strips in the urine sample. Any positive test will lead to immediate exclusion from the study. The urine sample, as well as the test strip, will be disposed of immediately on-site after testing. Documentation of the result is pseudonymised and will not contain any information for personal identification.

**2.4.2 Quantification of an effect of the protocol on the circadian system.** The experimental light exposure may affect the circadian system. However, indicators of the circadian systems are supposed to function as independent variable to predict visual outcomes. This may pose a problem of circularity, through which the effects of interest may be masked. The primary goal of the control sessions to estimate the effects of light exposure on circadian outcomes. During both control sessions, circadian outcomes described in Sect 2.4.7 will be measured.

To measure the effects of experimental light exposure on circadian outcomes, participants will visit the lab one week into the circadian stabilisation period, i.e. one week prior to the forced-desynchrony session, for the light exposure session. Saliva collection starts four hours prior to the participant's bedtime, will be carried out every 30 minutes until 1 hour after habitual bedtime. For more information on the post-processing, go to Sect 2.4.7. During this time, participants will complete similar tasks as in the forced-desynchrony. The visual experiments serve to imitate the light exposure that will be experienced during the protocol. Simultaneously, they are introduced and familiarised with the experiments that will be performed during the forced-desynchrony session, and will be equipped with the continuous glucose monitor the morning after.

Furthermore, participants will enter the lab on the day before the forced-desynchrony session starts to spend an adaptation night in the lab before the protocol starts. They will enter lab approximately five hours before their habitual bedtime, 13 hours before the forced-desynchrony session. They will stay under constant dim light throughout the evening preceding the adaptation night (dim light session). Data on salivary melatonin concentration and CBT will be collected throughout the evening. Comparing indicators of circadian phase measurements from the light exposure session and the evening dim light session, we will obtain an estimate of the potency of our stimulus protocol itself in affecting the circadian system.

**2.4.3 Circadian stabilisation and compliance.**

*Timing of homeostatic processes.* To stabilise their circadian system before entering the forced-desynchrony session, participants will start a two-week circadian stabilisation period. It is characterised by the maintenance of a strict 16:8-hour wake-sleep and light-dark schedule, and adherence to approximate meal times. Participants choose the sleep time based on their habitual sleep and wake time, and must then adhere to bed and wake times with a maximum deviation of 30 minutes.

*Caffeine, drug, alcohol and napping abstinence.* Caffeine modulates the circadian response to light [41]. To avoid possible influences, we ask participants to refrain from napping during daytime, as well as alcohol, recreational drug, and caffeine consumption seven days prior to lab visits.

*Daily questionnaire.* The 15-item consensus sleep diary serves as confirmation of sleep-/wake categorisation by the actigraph [42] and will be administered via REDCap. Participants will be asked to plan notifications on their personal smartphones in order to receive prompts to complete the diary in the morning within approximately 1 hour after awakening. Participants will be asked to record their mealtime. For this purpose, they will be asked to answer questions about their meals (e.g. timing, size) and take a picture to submit it to the app.

*Compliance in ambulatory assessments.* Compliance to the circadian stabilisation will be monitored with daily logs, the sleep diary and actimetry data. Participants will be instructed to wear a light-weight actiwatch (35 g) by (ActTrust2, Condor Instruments, São Paulo, Brazil),

on their non-dominant hand at all times throughout the 21 days of participation, except for when it could get wet or damaged. The device is equipped with a built-in accelerometer and light sensors. Activity and light exposure are measured in 1-min epochs. Light exposure is quantified in photopic illuminance and UV irradiance. Whether a participant is labelled awake or asleep depends on whether the activity exceeds a specified threshold for the activity counts. The data will be used to confirm compliance with circadian stabilisation, i.e. sleep and wake times, and may further be used in exploratory analysis evaluating relationship of main outcomes with regularity of light exposure (Light Regularity Index; LRI) and sleep (Sleep Regularity Index, SRI)[43].

*Metabolic activity.* We will monitor metabolic activity starting the morning after the light exposure session for a total of 14 days, thereby extending into the stabilisation period after the forced-desynchrony session. A continuous glucose monitoring (CGM) system (FreeStyle Libre 3, Abbott Laboratories, Alameda, California) is used to monitor participants' glucose profile. The small sensor is placed on the skin of the back of the upper arm, and measures the glucose concentration in the interstitial fluid (ISF). As blood glucose levels seem to rise in food anticipation and reflect meal-time history of multiple days, they will likely not be in line with the meal-timing imposed during the ultra-short sleep/wake rhythm [44]. Participants will not have access to their glucose readings during wear time, as feedback may interfere with natural eating patterns. Glucose levels are quantified in mg/dL.

**2.4.4 In-laboratory visits.** During the forced-desynchrony session, participants will repeatedly undergo a battery of visual, physiological and psychometric tests during wakefulness. The experiments that will be administered during one repeatable block are described in Fig 3 and detailed below.

*Compliance during in-laboratory visit.* The experimenter will continuously monitor compliance and the successful completion of experiments. Given the demanding nature of the protocol, there is tolerance for participant's failure to perform in individual tests. A block's data is considered for analysis as long as one test repetition within that block is successful. However, experimentation will proceed across all modalities using light-emitting devices, even if data collection issues occur in a single modality. These devices serve a dual purpose: not only do they measure visual outcomes, but they also provide consistent light exposure > 10 lux, which is important for maintaining comparability between participants. If the experimenter fails to collect data from both primary modalities – pupillometry and psychophysics – for more than one block due to any reason (e.g., non-compliance, fatigue, tiredness), the experimental session may be aborted. This is because the increased measurement intervals would render the data unsuitable for testing circadian hypotheses.

**2.4.5 Pupillometric assessment of photoreceptor mechanisms.**

*Maxwellian-view pupillometry.* The apparatus used for pupillometry is a custom-made six-primary binocular Maxwellian view light stimulation system [45]. The system is optimised to target different photoreceptors in isolation, including melanopsin. Light is produced by LEDs of different wavelengths (420, 450, 470, 520, 590, and 630 nm) with a bandwidth (FWHM) of approximately 20 nm each. The produced light stimulus is an annular field of 10–30° that is presented in monocular viewing conditions to the dominant eye. Participant movement will be limited using an elastic headband. The device measures pupil diameter over time based on video recordings of the respective eyes. Our main outcome is the PLR, i.e. pupil constriction upon monocular stimulation with visual stimuli of different spectral composition. Calibration of the stimulation device is done using a spectroradiometer (spectraval 1511, JETI, Jena, Germany) focused on a diffuser surface at the location of the entry pupil, and an optical

power and energy meter (PM100D, Thorlabs, Bergkirchen, Germany). There will be no individual calibration procedure. Calibration of the device and calculation of the employed light spectra is described below.

*Device calibration.* The devices were carefully calibrated beforehand. Here, we describe calibration and optimization procedures we conducted in order to ensure accurate measurements for a standard observer. Calibration was done once before data collection started. Individual calibration procedures for visual experiments are not feasible in the present study. Yet, it is possible to estimate the variability in the effective contrast. To calibrate the pupillometer, it was placed in the laboratory, which provides a stable environment during the day and during experimentation (no temperature or humidity fluctuations). For the calibration process, each LED was activated and measured on separately and at different intensity settings. As experimental setup, the light beam was projected onto a ground glass optical diffuser. This diffuser, with a 1-inch diameter, was made from N-BK7 glass and featured a 220 grit surface, making it ideal for effectively scattering light uniformly across the 350-700 nm wavelength range. The diffuser was mounted in front of the light source at the pupil plane of the participants. The distance between the light source and diffuser increased the effective surface of the light beam to about 2/3-inch diameter. The spectroradiometer was positioned close enough to the diffuser so that the effective surface area extended the receptive field of the Jeti spectroradiometer. Using the device's control software, we set LED intensities at 0%, 1%, 2%, and 5% steps. We then measured the remaining intensity spectrum in 5% increments, up to 10%, resulting in a total of 23 different intensity levels. We obtained spectral intensity distributions with a spectral resolution of 1 nm. We validated our spectral radiance measurements independently with a powermeter: we measured the power emitted at all the light intensities of each LED. An optical powermeter was connected via a cylinder-shaped applicator to the lenses of the device, in similar proximity as the pupil would be. This way, the light beam emitted from the device fell in the receptive field of the powermeter, while it was shielded from external light sources. The experimental light conditions were generated using *PySilSub*: as background, we specified all LEDs to be at half maximum intensity, and selected a constrained numerical optimization procedure. The software also requires a so-called calibration as input, which is with power-adjusted light spectra at different intensities generated by normalizing the measured spectra with the empirically measured power.

*Silent substitution stimuli.* Spectral composition of the light stimuli is varied to achieve silent substitution of the photoreceptor mechanisms of interest. Briefly, this means the activity of one or more photoreceptors is selectively modulated while the activation of other photoreceptors is kept constant [13,46–48]. While rods are saturated photopic light levels [49,50], stimuli will modulate melanopsin and cone activation for five conditions: (1) joint modulation of all photoreceptors (Light flux, melanopsin-containing ipRGCs, L-, M-, and S-cones) (2) joint modulation of all cones while keeping melanopsin activity constant (L+M+S) (3) selective modulation of melanopsin activation (Mel, constant of L, M, S activation) (4) selective modulation of S-cone activation (S) (5) selective modulation of the red-green colour channel activity (L–M). Comparing the four listed conditions against the Light flux condition, we will determine the unique circadian variation of the photoreceptor mechanisms.

The action spectra we used for the calculation of the silent substitution spectra stem from the default settings of the *PySilSub* package: the cone fundamentals of the macula from [58], and the melanopic and rhodopic action spectra [59] for a 32-year-old standard observer. We did not perform an age-adjustment considering different lens transmittance, when calculating the silent substitution spectra. We will simulate the response and report the residual variation, or deviation between obtained and target contrast, in the contrast of retinal mechanisms

and photoreceptors after silencing. We will disclose the effective precision of photoreceptor by calculating the so-called *contrast splatter* using the same method as [13].

*Experimental procedure.* During the forced-desynchrony session, every participant then completes 11 blocks, with a test and a test repetition in each block. Each test encompasses an adaptation period and 25 trials. Participants adapt to a photopic background (60 cd/m$^2$) for four minutes, to ensure rod saturation. The adaptation period is followed by 25 trials per test, where participants will view sinusoidal light modulations with a frequency of 0.4 Hz, interspersed with white baseline stimulation. 0.4 Hz was selected to measure pupil responses in a temporal regime in which all photoreceptors are likely responsive [13]. The experimental stimulus presentation takes place for five seconds, followed by a period of baseline stimulation with white light of 13 seconds, yields a total trial duration of 17 seconds. One test consists of 25 trials, five repetitions of each experimental condition; conditions are counterbalanced using an m-sequence. One test takes about 12–15 minutes, amounting to about 30 minutes per block for pupillometric assessment. A short break temporally separates the two tests.

*Control for rod intrusion.* As recent literature as shown that rods phototransduction may recover even at high light levels and based on temporal modulation [50,51], we will conduct a control experiment to test whether rods escape saturation. The control experiment will take place after data collection of the main study has been completed and will include a different set of participants than the participants in the main trial. Using pupillometry, we will evaluate the response to silent substitution stimuli targeting rods at light levels comparable to the planned experimental condition and with the same temporal frequencies. The results of this control study will be reported and discussed in the context of the results seen in the main study in subsequent publications.

### 2.4.6 Psychophysical assessment of photoreceptor mechanisms.

*Psychophysics testing with pupil relay system.* Psychophysical tests will be conducted using the Metropsis system (Cambridge Research Systems, Rochester, UK). Visual stimuli will be presented on a colour-calibrated three-primary 32-inch LCD with a 10-bit colour resolution and a refresh rate of 120 Hz. The field of view on the monitor was restricted to a circular blackout cutout approximately 38 cm in diameter that was placed in front of the monitor. The monitor was calibrated by the manufacturer at setup. While the photopic background ensures saturation of rods, the activation of melanopsin cannot be independently controlled. In all experiments, participants will view stimuli with their dominant eye through a pupil relay system: an artificial pupil restricts the pupil diameter to 1.5 mm. Image projection is done by relaying the image from the plane of the external pupil through an assembly of two lenses onto the plane of the participant's pupil. With this setup, the image is flipped horizontally, which we account for in the experimental setup. We correct for the participant's refractive error before starting the first experimental block. This is achieved by adjusting the distance between the two lenses, which shifts the focal plane to compensate for the refractive error of the human eye. By adjusting the focus, the image from the artificial pupil is brought into focus on the plane of the human pupil. This ensures a sharp image is formed without altering the magnification of the optical setup. This correction is maintained for all subsequent measurements. The experimenter controls the distance between the lenses. The participant is instructed to indicate when a change in distance causes the image to become blurred. The experimenter then adjusts the distance in the opposite direction until the participant indicates that the image is clear again. Participants' responses will be recorded on a response box (Black Box Toolkit Ltd., England). Two tasks will be carried out with this setup, a 10-axis Cambridge Colour Test (CCT) for screening purposes, and a temporal contrast sensitivity task (tCS task) for the perceptual threshold as main outcome, described in the following. Before the tasks, participants adapt to a neutral grey background at 60 cd/m$^2$ for one minute.

*Silent substitution of post-receptoral mechanisms.*   We will probe the sensitivity for luminance and of the colour-opponent processes by means of (1) joint modulation of all three cones to target the luminance channel (L+M+S*), (2) isolated modulation of S-cones keeping the luminance channel constant (S*), and (3) antagonistic modulation of L- and M-cones to target the red-green channel (L–M*). The asterisk in condition labels indicates the lack of control over melanopic stimulation with the RGB monitor compared to the 6-LED stimulation device employed during pupillometry, e.g. S* is the S-cone isolating condition on the RGB monitor, where melanopsin activation cannot be kept at a constant level, while S is the S-cone isolating condition fixing L-, M-, and melanopsin-activation while modulating S activity. Maximum contrasts are predefined by the given colour gamut of the monitor (L+M+S*: 100%, S*: 51%, and L–M*: 6%) and starting values are partly lower as a trade-off between experiment length and visibility of the stimuli (L+M+S*: 90%, S*: 35%, and L–M*: 6%). The accidental activation of melanopsin will not be monitored. Effects of ipRGCs on the psychophysical tasks performance are assumed to be minimal.

*Post-receptoral temporal contrast sensitivity.*   In the two-alternative forced choice (2AFC) tasks, stimulus contrast varies according to a staircase method, approaching the participant's contrast threshold in relative steps. Contrast sensitivity is calculated from the threshold values. Within one test of the tCS task, different temporal frequencies, 2 and 8 Hz, are presented in interleaved order for one condition or photoreceptor mechanism. Participants determine whether the stimulus appears right or left from the centre of the screen with a left or right button press. Location varies randomly. After completion of three conditions, we will reiterate over the conditions twice, yielding nine tests per block. During the light exposure session (Sect 2.4.2), the participant will learn the task procedure, completing all 9 tests to achieve comparable light exposure levels. Tests are expected to last longer due to unfamiliarity in the light exposure session.

**2.4.7 Circadian phase measurements**   Circadian phase will be determined by body temperature, salivary melatonin concentration, and DLMO.

*Telemetric core body temperature.*   Temperature in the gastrointestinal tract will be measured through a telemetric pill. Data will be sent from ingestible pills from the e-Celsius® Performance System 261 (BodyCap, Hérouville Saint-Clair, France) with a sampling rate of 2 Hz, providing measurements every 30 seconds. Device connects to an external device when in proximity and sends data via RFID. The first pill will be administered in the beginning of each session (light exposure session, dim light session, forced-desynchrony session) and another one 20 hours into the forced-desynchrony protocol, as the first pill will be excreted in the stool after 24–48 hours. The device is CE-certified. As the device is not MRI safe, participants will receive a wristband indicating that they cannot enter an MRI scanner. Temperature data will be used to estimate the person-specific circadian period $\tau$ with available data dim light session, the subsequent adaptation night and the forced-desynchrony session as described in Sect 2.6.1.

*Salivary hormone concentrations.*   Melatonin and cortisol concentration will be obtained using from saliva samples that are sampled approximately every 45 minutes throughout the forced-desynchrony, except during sleep periods. Saliva will be sampled approximately every 30 minutes starting 4 hours prior to bedtime until habitual bedtime. Salivette® collection devices were chosen for saliva samples and bioassaying for the extraction of hormone concentration saliva collection. In brief, participants chew on a cotton swab for 5 minutes and place it into a plastic collection tube. The Salivette® will then be centrifuged for 3 minutes at 3,000 rpm and frozen at -20° for temporary storage. Biological samples will be collected initially in the laboratory and then sent to external laboratory companies for further processing. Saliva

samples of at least 0.5 mL will be taken from the participants using Salivettes (Sarstedt, Nürnberg, Germany). Melatonin and cortisol will be determined by ELISA (CRTN-96, NovoLytiX GmbH, Switzerland, as commissioned work. The analyses have the typical specifications: limit of quantification 0.3 – 30 ng/mL, detection limit $\leq$ 0.3% ng/mL, intra-assay precision $\leq$ 6%, and mean inter-assay precision $\leq$ 10%. All samples will be destroyed after completion of the hormone analyses. The salivary melatonin concentration during forced-desynchrony is used to identify the participant-specific circadian period. Dim light melatonin onset (DLMO) will be determined for both evenings, using the hockey stick algorithm [52].

**2.4.8 Homeostatic processes.** We will collect data from multiple modalities to accurately assess the homeostatic state.

*Subjective sleepiness.* As measure of subjective fatigue, participants will be asked to report their sleepiness via the Karolinska Sleepiness Scale (KSS) at every measurement block.

*Distal-proximal temperature gradient.* Wireless iButton devices will be used to monitor skin body temperature at two pre-defined locations on the skin, one proximal, e.g. near the neck, and one distal, e.g. on one ankle. The thermometry devices will also be attached to the skin. From the sensor readings, the temperature gradient between the distal and proximal skin (DPG) will be calculated, as it has been shown to vary with sleep propensity [53]. Sampling frequency for skin temperature is 2 Hz. Information derived from skin temperature may feed into exploratory analyses, statistically separating effects of the circadian rhythm and homeostatic processes or the sleep-wake rhythm.

*Cardiovascular and respiratory monitoring.* For blood pressure monitoring during the experimental sessions, we will employ the ABPMpro device (SOMNOmedics AG, Randersacker, Germany). Blood pressure measurements will be taken twice per measurement block. The ABPMpro can be extended by a 3 EGC sensor (Holter). We will attach the surface electrodes to the participant's chest and ribs for impedance cardiography throughout the 40 hours forced-desynchrony. We may derive outcomes such as heart rate, heart rate variability, and breathing frequency from measurements, to evaluate activity of the autonomous nervous system in exploratory analyses.

*Alertness and Vigilance.* In every block, participants will complete a 10-minute visual psychomotor vigilance test. The test will be delivered on the PVT-192 Psychomotor Vigilance Task Monitor (Ambulatory Monitoring, Inc., Ardsley, USA; [54]) and serves as a measure of alertness, vigilance or conversely, sleepiness. As part of the test, participants are tracking a visual stimulus, a digit counter that appears at random on the device display and starts counting up in milliseconds. The display acts as both, the stimulus and the performance feedback display. Participants have to respond as fast as possible to the stimulus with a button press using the thumb of their dominant hand. Time between stimulus presentation and response constitutes the participant's response time if they are below or equal 500 ms. Median reaction time, lapse probability (reaction time > 500 ms) per session, and mean of fastest 10% reaction times are used as main outcome measures of alertness.

**2.4.9 Ocular structure and function.** All measurements of ocular physiology will be performed on the left and the right eye. Data on various aspects starting from light incidence at the retina to structure of the retinal layers will be collected. All applied procedures are standard procedures that are routinely carried out during a visit to the ophthalmologist. The ophthalmologist of the team will review concerning results and write a physician's letter in case of incidental findings.

*Photography of fundus and anterior segment.* During the first session, coloured images of the anterior segment (including iris) and the fundus will be made.

*Optical toherence tomography measurements.* We will repeatedly capture structural images of the retina and the cornea (without concomitant fundus photography due to light exposure),

to obtain estimates of corneal and macular thickness using time-domain optical coherence tomography system (OCT-2000, Topcon Corporation, Tokyo, Japan).

*Intra-ocular pressure.* For measurements of intra-ocular pressure (IOP), we will use a Non-Contact Tonometry (NT-2000 NC Tonometry, NIDEK CO., LTD., Japan). The outcome variable for one measurement block is the arithmetic mean of three estimates, quantified in millimetre of mercury. Non-contact tonometry involves administration of an air-puff to the eyes, which may lead to a temporary sensation of dryness or discomfort in the eyes.

**2.4.10 Subjective well-being.** Participants stay under laboratory conditions for several days. This touches upon most aspects of everyday like, for example, eating habits during this period will be monitored. However, participants are allowed to have a minimum of 30 minutes of their own time during each awake period. Additionally, their emotional state is closely tracked with mood questionnaire (I-PANAS-SF) [55].

*Enrichment.* During the experimental sessions, participants will not be able to use their phones, or any other device indicating time of day, nor will they be allowed to receive visitors. Relatives or close ones may be given a phone number to call the study team on-site in the event of an emergency. For enrichment during their stay, they may bring their own media or reading material, or may choose from pre-selected music, podcasts, or audiobooks.

*Physical needs.* During the laboratory visits, participants will have access to water ad libitum to maintain hydration. For their nutritional needs, they will be provided with shakes containing a balanced composition of macronutrients (carbohydrates, protein, fats, and fibre) and essential micronutrients. Shake intake will be scheduled exclusively during wakefulness, and participants will be instructed to consume the shakes at a frequency consistent with their usual number of main meals per day (typically between two and four meal times). To accurately determine caloric intake during the 40-hour forced-desynchrony session, each participant's base metabolic rate (BMR) will be calculated using the Mifflin-St. Jeor equation, which accounts for sex, body weight, and height [56]. The BMR will then be adjusted for a sedentary lifestyle using a correction factor of 1.2, irrespective of participants' habitual levels of physical activity.

## 2.5 Outcomes

In accordance with the different sets of hypotheses, the primary outcomes refer to the image-forming pathway enabling perception, and the non-image forming pathway responsible for non-image forming visual functions.

**2.5.1 Image-forming visual function.** The image-forming pathway is investigated using psychophysical tests, testing of contrast thresholds. During the tCS task, the contrast threshold is determined; contrast is measured in percent. We will average the contrast threshold block-wise across test repetitions, separate for each condition. A linear mixed model will be used to investigate hypotheses H2a to H2c. Our primary outcome variable is the point estimate for the temporal contrast sensitivity in each test for a specific post-receptoral mechanism and a specific frequency. The temporal contrast sensitivity is the inverse of the contrast threshold determined by a two-up-one-down staircase method.

**2.5.2 Non-image-forming visual function.** For the non-image forming pathway, we test the PLR using pupillometry. There are various metrics to quantify pupil kinetics, but the maximum pupil constriction is particularly well-suited to capture the rapid photoreceptor responses of the cones. The primary outcome variable is the relative pupil constriction $R$ in response to a receptor- or mechanism-isolating stimulus. To obtain a measure of relative pupil constriction, we will obtain a maximum pupil constriction amplitude per trial from the pupil trace. Maximum pupil constriction is defined by the maximum difference between

pupil diameter one second prior to stimulus onset and the pupil diameter during the 5-second stimulus presentation. We will calculate the average pupil response amplitude across trials and test repetitions for each condition. The final outcome measure is calculated by normalising the average pupil response amplitude in response to a specific condition against the average response amplitude to the combined modulation of all cones and melanopsin at equal contrast "(Light flux stimulation)", as specified in eq. (1).

$$R_{\text{condition}} = \frac{A_{\text{condition}}}{A_{\text{all}}} \tag{1}$$

We use the PLR to refer to the ratio $R$ of response amplitudes. To investigate hypotheses H1a to H1d, we will use statistical modelling linear mixed effects as detailed in eq. (4).

## 2.6 Statistical analysis

**2.6.1 Estimation of circadian period.**   We will use a symmetric sinusoidal regression model to estimate the fundamental period $\tau$ of the circadian pacemaker. The model will be fitted to salivary melatonin data, which serves as a proxy for the centrally governed circadian rhythm. The formulation of the regression model is as follows:

$$s_{\tau_j}(t) = m_0 + a \cdot \sin\left(\frac{2\pi t}{\tau_j}\right) + b \cdot \cos\left(\frac{2\pi t}{\tau_j}\right) + \varepsilon \tag{2}$$

where $t$ is the time passed since beginning of the protocol, measured is the average CBT. The three parameters to be fitted for each individual participant are the intercept or MESOR (Midline Statistic Of Rhythm) $m_0$, and $a$ and $b$ which are the weights for the sine and cosine component, respectively. We will estimate multiple models, iterating over values of $\tau \in [20; 28]$, and we will select $\tau$ with the best fit statistics *AIC*. The point estimate of $\tau$ will be propagated into the confirmatory models testing for a circadian rhythm in visual functions.

**2.6.2 Confirmatory analysis.**   For our confirmatory hypotheses, we will examine two types of models. One model will test confirmatory hypotheses H1a–H1d investigating circadian rhythms in the PLR, and one model will be used to test hypotheses H2a–H2c concerned with circadian rhythms in the psychophysical thresholds. To test our hypotheses, we will always compare a full model incorporating a data generating circadian process, against a null model assuming a constant level in the dependent variable.

To test our hypotheses about the PLR, we will examine the ratio $R$ of response amplitudes as a function of a participant's circadian rhythm by calculating the Bayes factor of the comparison between a full model and the null model. The circadian rhythm is modelled with a sine-cosine parameterisation using the point estimate $\tau$ from eq. (2). The null model without the circadian components is defined, as

$$R_j(t) = m_0 + u_{0j} + \varepsilon \tag{3}$$

while the full model with the circadian components is defined as

$$R_{\tau_j}(t) = [m_0 + u_{0j}] + [a + u_{aj}] \cdot \alpha_{\tau_j}(t) + [b + u_{bj}] \cdot \beta_{\tau_j}(t) + \varepsilon \tag{4}$$

where $m_0$ is the overall intercept or MESOR (Midline Statistic Of Rhythm), $u_0$ is the person-specific random intercept. The sine and cosine components $\alpha$ and $\beta$ together represent amplitude and phase of the circadian rhythm of a person $j$ with a person-specific period $\tau$. They

vary over time $t$ with fixed effects $a$ and $b$ and random participant-specific effects $u_a$ and $u_b$, with $u_a \sim \mathcal{N}(0, \sigma_{u_a}^2)$, and $u_b \sim \mathcal{N}(0, \sigma_{u_b}^2)$. $\varepsilon$ is the error term associated with the response of a person $j$ at a specific time $t$.

Similarly, circadian modulation of psychophysical performance is assessed by testing for a circadian component in temporal contrast sensitivity $T$. The null model $T(t)$ and the full model $T_{\tau_j}(t)$ are defined as in eqs (3) and (4) above. Model comparisons are conducted separately for low (2 Hz) and high (8 Hz) frequencies. While sensitivity differences between post-receptoral mechanisms are well-documented, there is currently no evidence to suggest frequency-dependent differences in circadian modulation. For the sake of parsimony, if results between low and high frequencies or across post-receptoral mechanisms differ, we will perform exploratory analyses. We will not enter sex, age, or demographic variables in our confirmatory analysis, as we do not see sufficient empirical evidence for confounding effects.

We will sample from the posterior distributions to obtain parameter estimates and derive amplitude ($A$) and phase angle ($\phi$) of the proposed underlying circadian rhythm with

$$A = \sqrt{a^2 + b^2} \tag{5}$$

and

$$\phi = \begin{cases} \arctan(\frac{-b}{a}), & a > 0 \\ \arctan(\frac{-b}{a}) + \pi, & a < 0 \text{ and } b \geq 0 \\ \arctan(\frac{-b}{a}) - \pi, & a < 0 \text{ and } b < 0 \\ -\frac{\pi}{2}, & a = 0 \text{ and } b > 0 \\ +\frac{\pi}{2}, & a = 0 \text{ and } b < 0 \end{cases} \tag{6}$$

**2.6.3 Exploratory analysis.** Effects found in the previously mentioned models will be further explored in additional analyses. Building upon the results from hypotheses H1a–H1d and H2a–H2c, we may further explore circadian effects specific in temporal frequencies for specific post-receptoral mechanisms. Additionally, we may confirm circadian effects in ocular structures and physiological parameters. Trajectories of ocular structures or summary statistics of physiological parameters may be included in post-hoc analyses as confounding factors in the model described in eq. (4) for hypotheses H1a–H1d, and for hypotheses H2a–H2c.

## 2.7 Ethical approval

The protocol has been reviewed and approved by the Medical Ethics Committee of the Technical University of Munich (2023-369-S-SB). All research will be conducted in accordance with the Declaration of Helsinki.

# 3 Discussion

## 3.1 Strengths and weaknesses of the protocol

Our protocol is uniquely geared to measuring circadian variations of retinal mechanisms while minimizing homeostatic effects. In evaluating our protocol, we highlight several points.

*Selection of pupillary response metrics and stimulation.* There are various measures of pupil kinetics that vary in their contribution of different photoreceptor classes: commonly, one differentiates between metrics of the pupillary light response and the post-illumination pupil

response (PIPR) [57]. The contribution of different photoreceptor classes to pupil dynamics varies temporally and depending on stimulus design. The PLR is more focused on the initial constriction, while the PIPR is focused on the time period after offset of a light stimulus, including the redilatation phase. The PIPR measures the response to monochromatic light stimuli to no or a dim background, and offers common metrics to investigate dynamics of melanopsin-mediated visual functions. PIPR outcomes do allow to contextualize results better in previous literature. However, we have chosen the PLR because the experimental design associated with PIPR outcomes includes periods of dark adaptation and relies on long trials, unspecific photoreceptor activation, and little to no background illumination. To conclude, it forgoes the advantages of silent substitution, such as efficiency and specificity, for less complexity in experimental design.

*Known limitations in silent substitution.* The silent-substitution stimuli we use are nominally designed to stimulate specific photoreceptor classes. The precise contrast seen by the photoreceptor classes in the actual stimulus conditions can vary due to individual differences, which is difficult to characterise for an individual. As a consequence, our stimuli may contain some small residual contrast on the unstimulated photoreceptor classes. This is a well-known problem [47] but does not challenge the validity of silent-substitution measurements at large contrasts.

*Artificial laboratory conditions.* Due to the controlled nature of the experiment, participants are placed in an artificial environment, unable to fully represent real-world conditions. In particular, the experimental light conditions may have an impact on psychological variables in such a way that behaviour diverges from behaviour in natural settings. This is a limitation of any laboratory study, but a precondition for experimentally manipulating the mechanisms under investigation.

*Sample considerations.* In our study, we aim to recruit a total of twelve participants. As our study follows a strong within-subjects design, we believe that this sample size is appropriate to capture within-participant circadian influences on the outcome measures. Importantly, unlike many chronobiological studies, the present study is unique in that it strives to have a representative sample regarding distribution of sex (50% female participants). Due to unknown variations introduced by hormonal fluctuations, data required for comparisons will be collected at similar stages in the menstrual cycle, specifically during the mid-follicular phase (see Sect 2.4.2).

## 3.2 Dissemination

We are dedicated to ensuring the robustness and transparency of the findings of this research, and its dissemination. First, we will present our results at relevant academic conferences, fostering discussion and feedback to ensure adequate data processing. All resources, including code, data, recruitment materials will be made openly available on GitHub in order to promote transparency and reproducibility. Finally, where applicable, we are committed to engage with the public to convey the importance and applicability of the presented research, as well as to foster a better understanding of the underlying biological phenomena, such as circadian rhythms, sleep-wake behaviour and visual performance.

## 3.3 Protocol amendments, deviations and termination

In the publications describing the results arising from this protocol, any termination of the protocol will be described transparently and published for the scientific community, openly discussing conditions and feasibility of the presented study protocol, as well as relevant learnings. Any deviations from or ad-hoc amendments of the protocol will be described, along

with the rationale. At the data analysis stage, all available data will be utilized. In the expected rare occurrence of termination by the participant, we will describe circumstances and if available reasons for termination, and where possible and in line with the inclusion criteria, we will include partial data in our analysis.

## Author contributions

**Conceptualization:** Hannah Sophie Heinrichs, Manuel Spitschan.

**Funding acquisition:** Manuel Spitschan.

**Investigation:** Hannah Sophie Heinrichs, Manuel Spitschan.

**Methodology:** Hannah Sophie Heinrichs, Manuel Spitschan.

**Project administration:** Hannah Sophie Heinrichs, Manuel Spitschan.

**Software:** Hannah Sophie Heinrichs.

**Supervision:** Manuel Spitschan.

**Validation:** Hannah Sophie Heinrichs.

**Visualization:** Hannah Sophie Heinrichs.

**Writing – original draft:** Hannah Sophie Heinrichs.

**Writing – review & editing:** Hannah Sophie Heinrichs, Manuel Spitschan.

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
