## [Decision Letter · Decision Letter 0]

11 Jun 2024

PONE-D-24-07303Within-subjects ultra-short sleep-wake protocol for characterising circadian variations in retinal functionPLOS ONE

Dear Dr. Spitschan,

Thank you for submitting your manuscript to PLOS ONE. After careful consideration, we feel that it has merit but does not fully meet PLOS ONE’s publication criteria as it currently stands. Therefore, we invite you to submit a revised version of the manuscript that addresses the points raised during the review process.

We look forward to receiving your revised manuscript.

Kind regards,

Daniel Joyce

Academic Editor

PLOS ONE

Additional Editor Comments:

Thank you for your submission. Your manuscript has been reviewed and the reviewers are enthusiastic about the manuscript. However, they also raise important points, especially around methodology, rationale, and terminology. I ask that you please give these careful consideration and revise the manuscript accordingly.

Reviewers' comments:

Reviewer's Responses to Questions

**Comments to the Author**

1. Does the manuscript provide a valid rationale for the proposed study, with clearly identified and justified research questions?

Reviewer #1: Yes

Reviewer #2: Partly

2. Is the protocol technically sound and planned in a manner that will lead to a meaningful outcome and allow testing the stated hypotheses?

Reviewer #1: Partly

Reviewer #2: Partly

3. Is the methodology feasible and described in sufficient detail to allow the work to be replicable?

Reviewer #1: Yes

Reviewer #2: No

4. Have the authors described where all data underlying the findings will be made available when the study is complete?

Reviewer #1: Yes

Reviewer #2: Yes

5. Is the manuscript presented in an intelligible fashion and written in standard English?

Reviewer #1: Yes

Reviewer #2: No

6. Review Comments to the Author

You may also provide optional suggestions and comments to authors that they might find helpful in planning their study.

Reviewer #1: Revision of “Within-subjects ultra-short sleep-wake protocol for characterising circadian variations in retinal function”

The study protocol from Heinrichs and Spitschan is carefully well-designed. It deals with a major question in the intersection of vision and circadian science, which is whether circadian variation in human physiology has a consequence in early light information processing. They focus on vision sensitivity and pupillometry to study this potential effect.

In the following text, I include my comments on the manuscript starting with the major concerns.

The hypotheses of this study are based on the selective and independent stimulation of each each post-receptoral pathway (L+M+S, L-M, and S) for both, pupillometry and psychophysics. To achieve this selectivity the authors will use the silent substitution method. However, this method can be affected by individual variability. The authors recognize this weakness (Line 717) but they claim that it shouldn’t be a problem with high-contrast stimulation. With the devices that they are going to use, they might achieve a good contrast for L+M+S and possibly S, however, L-M contrast is usually very low with any device especially when controlling melanopsin. The authors should include an individual calibration procedure and report what are the possible contrasts for each stimulation condition with their devices. This is very important since the main outcomes of the study are based on this selectivity.

The hypotheses are a little poorly detailed. Based on the literature the authors might want to include what they expect. For example, Zele and colleagues found that melanopsin-driven contribution is higher in afternoon hours (Zele et al, Plos One, 2011). Do the authors think that something similar or contrary will be expected in their study?

In general, the captions’ figures need to contain a more detailed description.

Although the tradition in vision science is that rods are saturated in photopic light levels as the authors pointed out (Line 405), there is new evidence that rods might signal at high light levels (see Uprety et al, iScience, 2022). I think it is worth conducting a control experiment to discard rod intrusion in the planned experimental conditions or to better explain why rods would not affect the results of the planned study.

The effect of tiredness in psychophysical tasks and Pupillary recordings is well known. The authors have included an alertness and vigilance procedure to control this effect. However, I wonder what would be the exclusion procedure in this case. If the participants fail this test, then the complete measurement is excluded? I guess that the last sessions might be very demanding. The authors might want to explain a little bit more this procedure.

Section 1.4: Modulation of the cone-opponent process L-M in pupillometry. I wonder If the authors think that this hypothesis can provide information about the M-cone out-of-phase response found by (Woelders et al, PNAS, 2019, and Murray et al, JOSA A, 2019), which is not in agreement with a previous study (Barrionuevo and Cao, JoV, 2016) and the physiological evidence provided by Dacey and colleagues (Nature, 2005)?

Line 57-58: Maybe I’m not understanding well but it seems like a circular rationale to me.

Fig. 1: Is actigraphy conducted for 20 days, 21 days, or more (considering the figure)?

Line 300-302: glaucoma may cause disruptions to sleep (Gracitelli et al, IOVS, 2014) so the authors might want to explain more about it and the reason for this exclusion criteria.

Line 505: type “as” should be “has”

Reviewer #2: 1. Rationale: The umbrella study question is fantastic, and very timely. The one area that needs more justification is using the silent substitution method. Until that is the dominant method, and is validated against other more commonly employed methods (which arguably could be done here), it needs additional justification. Add rationale for light exposure midway through the protocol. Is this expected to shift the clock or not, and what about it's timing is intentional and why? Specifically, a clearer description on how the data from this visit will be compared with the FD protocol data i.e., how will this visit inform whether the tasks in the FD elicited a phase shift. Also, it is likely some measurement, particular phase markers, will have been influenced by the first week of the stabilization period. Please include a rationale for including this study visit at this particular timepoint (e.g., halfway through the stabilization) and not at a more representative baseline. Please also clarify whether this light exposure session mimics one cycle of tasks completed during the FD and if these will be conducted under the same lighting conditions as the FD.

2. Protocol: Please justify how sleep homeostasis will not vary across the 40-hour protocol so as to remove it's effect and allow the circadian effect to be disentangled. If it begins upon wake, process S will be low and sleep may not occur in the first few sleep opportunities, although process S will accumulate as the morning process C increase in alertness rises. If sleep cannot occur until the 3rd or 4th sleep opportunity, please explain how process S will be kept stable so as to isolate process C.

3. Please explain how the PLR will address non-image forming photoreception, as PIPR methods are more typically used for this with post-illumination measurement. Please provide additional justifications for pupillometry methodological decisions. First, it is unclear why only acute pupillary responses are being investigated given the proposed ability to isolate and capture melanopsin-driven responses to light. Given recent interest in the role of the melanopsin system in circadian photoentrainment, it seems remiss to not measure the post illumination responses of the melanopsin system to light which has been used more than the silent substitution methods. At the very least, both could be included since the pupillometry is already incorporated. Further, while standardizing entrance pupil diameter constrains retinal irradiance, it also removes potentially meaningful individual differences in responses to light. Please include a justification of this methodological decision and the consequences of not using pharmacological dilation and consensual response or Maxwellian view, for example. Finally, please report how dominant eye will be determined and why this is necessary.

4. Language - there are a few typos and confusing sentences that can be easily edited. These include "modulation of alertness of light" by which the authors might mean "modulation by light of alertness." The sentence "Studies investigating how the 58 pupillary light reflex, which reflects the response of all photoreceptors to light [12,13], have revealed that there are a time-of-day differences in melanopsin-mediated pupil responses [14,15], indicating that at least some variability originates in the retina" could be revised. There is a typo somewhere in "...and to create a regular cycles of sleep and wakefulness." An exhaustive list of possible edits hasn't been detailed here, so please review and edit carefully to ensure clarity.

5. In addition, the PLR and PIPR are distinct, and driven by different photoreceptor classes. It is not clear here how the time at which pupil diameter is measured correlates with the photoreceptors that are signaling, and how the PIPR differs in it's ability to reflect ipRGCs. Naturally, the rods and cones project to ipRGCs, but only ipRGCs are signaling in the post-illumination period. Until silent substitution measures are validated and standardized, including the PIPR allows researchers to balance the pros and cons of both.

6. Please specify inclusionary/exclusionary SRI cutoffs. In the metabolic processes section: 14 days should be after light exposure session through 1 week after FD based on study protocol diagram not until after the FD session. It may also be worthwhile to further explain how light history information from the circadian stabilization phase will be used to inform the FD pupillometry data.

7. Please more clearly lay out the importance and difference of the contrast sensitivity and silent substitution methods, and what they measure that is relevant to sleep and circadian rhythms. Visual function is not thought to vary, but perhaps this is of interest somehow?

8. The terms used in the field typically segregate into PLR for pupil light reflex, first reported by Lowenfeld in 1999, and PIPR for post-illumination pupil response, described in 2007 by Gamlin. If this is not the right terminology to be used, please clarify for readers so that the different goals of the PLR and PIPR are not confused and are more clear. Given that the PLR is often in response to light but also images or sound, I think it is too broad to be used to describe ipRGC driven repsonses to chromatic stimuli, and could be confused. How will you distinguish this? Is there a reason the post-illumination period is not preferred?

9. Adding the PIPR to the protocol would be a great service to the field.

10. Please explain why exactlty the Zele and Münch papers on diurnal variation in PIPR from about 2009 are not sufficient for your purposes, and specifically what they might have misrepresented by not employing FD measures. Is process S thought to affect the PIPR? I sleep homeostasis known to impact ipRGCs? Theoretically, is there a rationale for why the Zele and Münch papers are unlikely to reflect the daily variation in ipRGC responsivity?

9.

7. PLOS authors have the option to publish the peer review history of their article (what does this mean?). If published, this will include your full peer review and any attached files.

Reviewer #1: No

Reviewer #2: No

---

## [Author Response · Author response to Decision Letter 1]

20 Sep 2024

The reviewer comments are addressed in the "response to reviewers" file.

---

## [Decision Letter · Decision Letter 1]

20 Dec 2024

Within-subjects ultra-short sleep-wake protocol for characterising circadian variations in retinal function

PONE-D-24-07303R1

Dear Dr. Spitschan,

We’re pleased to inform you that your manuscript has been judged scientifically suitable for publication and will be formally accepted for publication once it meets all outstanding technical requirements.

Kind regards,

Daniel Joyce

Academic Editor

PLOS ONE

Additional Editor Comments (optional):

Thank you for addressing the reviewer's comments.

Reviewers' comments:

Reviewer's Responses to Questions

**Comments to the Author**

1. Does the manuscript provide a valid rationale for the proposed study, with clearly identified and justified research questions?

Reviewer #2: Yes

2. Is the protocol technically sound and planned in a manner that will lead to a meaningful outcome and allow testing the stated hypotheses?

Reviewer #2: Yes

3. Is the methodology feasible and described in sufficient detail to allow the work to be replicable?

Reviewer #2: Yes

4. Have the authors described where all data underlying the findings will be made available when the study is complete?

Reviewer #2: Yes

5. Is the manuscript presented in an intelligible fashion and written in standard English?

Reviewer #2: Yes

6. Review Comments to the Author

You may also provide optional suggestions and comments to authors that they might find helpful in planning their study.

Reviewer #2: The authors have responded to all of the concerns. since this is a report of a protocol that will be done and reported separately, some of the challenges like validation of the silent substitution methods for each participant can also be addressed then. But it will likely be an important consideration for the validity of the results. It would also still be possible to add in a PIPR protocol if the study has 18 minutes to spare. It would even be possible to do one PIPR trial within 13 minutes, with 11 min dark adaptation and 2 minutes for a PIPR to blue light. This would add to the informativeness of the work and could convince others to use silent substitution.

7. PLOS authors have the option to publish the peer review history of their article (what does this mean?). If published, this will include your full peer review and any attached files.

Reviewer #2: No

---

## [Editor Report · Acceptance letter]

PONE-D-24-07303R1

PLOS ONE

Dear Dr. Spitschan,

I'm pleased to inform you that your manuscript has been deemed suitable for publication in PLOS ONE. Congratulations! Your manuscript is now being handed over to our production team.

Kind regards,

on behalf of

Dr. Daniel Joyce

Academic Editor

PLOS ONE